# Authentication of Iranian Saffron (*Crocus sativus*) Using Stable Isotopes δ^13^C and δ^2^H and Metabolites Quantification

**DOI:** 10.3390/molecules27206801

**Published:** 2022-10-11

**Authors:** Benjamin Moras, Camille Pouchieu, David Gaudout, Stéphane Rey, Anthony Anchisi, Xavier Saupin, Patrick Jame

**Affiliations:** 1Activ’Inside, 12 Route de Beroy, ZA Grand Cazau, 33750 Beychac et Caillau, France; 2UMR5280 CNRS, Institut des Sciences Analytiques, Université de Lyon, 69100 Villeurbanne, France

**Keywords:** saffron, authentication, adulteration, stable isotopes analysis, crocins, safranal

## Abstract

Saffron is a very high value-added ingredient used in the food supplement market and contains a high level of safranal. Adding synthetic safranal to saffron, which is significantly cheaper, and falsifying the origin of saffron may represent recurrent fraud. Saffron from different countries was analyzed to determine the stable isotope ratios δ^13^C and δ^2^H from safranal by gas chromatography coupled with isotope-ratio mass spectrometry (GC-C/P-IRMS) and the concentration of saffron metabolites with ultra-high performance liquid chromatography coupled with diode array detector (UHPLC-DAD). The isotopic analysis highlighted a higher ratio of δ^2^H in synthetic safranal than in natural safranal; the mean values were 36‰ (+/− 40) and −210‰ (+/− 35), respectively. The δ^13^C between Iranian, Spanish and other saffron was significantly different and represents median values of −28.62‰, −30.12‰ and −30.70‰, respectively. Moreover, linear and quadratic discriminant analyses (LDA and QDA) were computed using the two isotope ratios of safranal and the saffron metabolites. A first QDA showed that trans-crocetin and the δ^13^C of safranal, picrocrocin, and crocin C3 concentrations clearly differentiated Iranian saffron from other origins. A second model identified δ^13^C, trans-crocetin, crocin C2, crocin C3, and picrocrocin as good predictors to discriminate saffron samples from Iran, Spain, or other origins, with a total ability score classification matrix of 100% and a prediction matrix of 82.5%. This combined approach may be a useful tool to authenticate the origin of unknown saffron.

## 1. Introduction

Saffron refers to the stigmas of *Crocus sativus* and is traditionally used in cooking, infusion or in food products. Its earliest representation appeared 4000 years ago in some craft products of the Minoan civilization on Crete Island [1]. Iran is currently the main producer, with approximately 90% of world production, which was estimated at 418 tons in 2020 [2]. The other main producers are India, Greece, Afghanistan, Morocco, Spain, Italy, and China [3]. Spain is the first saffron importer. Export and import trading data, compared to production, showed that Spain creates more value-added products in the saffron industry than other countries [4]. Saffron from Iran could be traditionally declined in different grades; the Sargol grade corresponds to the top of the red stigma, which allows having the high content in crocins compounds that gives the saffron its characteristic color [5].

Some health and well-being applications have been demonstrated in the literature. Health benefits of saffron have been suggested for the treatment of anxiety disorders [6,7,8], premenstrual syndrome [9], neurodegenerative retinal diseases [10], erectile disorders [11], and for mental health [12,13,14,15]. The saffron market for food supplements is constantly growing, with hydroalcoholic extracts marketed in capsules, tablets, or other galenic formulations. The recommended intake is conventionally based on hydroalcoholic saffron extract powder, standardized at 2% safranal by a UV-spectrometric method. Safranal is the major compound of the aromatic constituents, reflecting a key element of saffron quality. Safranal content could be influenced by traditional or modern drying method and consequently its content could depend on its origin [16]. This molecule is also of growing interest playing a key role in the prevention of mood disorders [17].

Since saffron is the most expensive spice in the world, the saffron trade is often affected by fraud. Therefore, the authentication of the raw materials or extracts is crucial, especially for the food industry or the dietary supplement market using powder as the main form of trading.

Firstly, traditional fraud consists of fully or partially replacing the saffron by another plant, such as other crocus, *Carthamus tinctorius*, or *Turmeric longa* [18], easily detectable by saffron experts. However, saffron powder makes the visual identification difficult. In this case, the microscopy techniques allow checking the presence of characteristic structures (e.g., pollen). Microscopic analysis and spectrophotometric measurement proposed in ISO 3632 are conventionally used to authenticate *Crocus sativus* and to authenticate and estimate its quality (categories I to III). DNA analysis could be complementary performed to confirm identity [19], but it does not detect the addition of a DNA-free product (e.g., dyes, crocin extract from other origins).

Secondly, the addition of other parts of *Crocus sativus* is also a possible fraud. Recently, a metabolomic approach by UHPLC/QTOF-MS demonstrated the potential of phenolics screening, particularly anthocyanins and glycosidic flavonols, as the best markers of this type of adulteration [20]. The presence of these compounds could be linked to the proportion of basal section of the flower. The second level of fraud consists of introducing natural or synthetic dyes on whole or ground stigmas of poor quality. These dyes, such as tartrazine, ponceau 4R, Sudan, bixin, yellow and orange dyes, can be detected by HPLC methods, also described in ISO3632 [21].

Thirdly, there are more sophisticated frauds, including natural compounds of saffron from other plants. Crocins from Gardenia jasminoide, which is cheaper than saffron, could be added to saffron powder. A powerful ^1^H NMR technique identifying adulterated saffron with 20% gardenia or other plant adulterants has been proposed [22]. The UHPLC method can also detect the presence of specific iridoids from gardenia [23] or the addition of gardenia crocin extract until 1% *w*/*w* [24].

Finally, safranal, the main volatile compound of saffron, guarantees organoleptic quality. Since it has been demonstrated that the ISO method is a source of significant overestimation error [25], safranal should be quantified by HPLC or GC techniques. Adulteration with synthetic safranal could be an insidious fraudulent practice. A small spiking of synthetic safranal added to powdered stigmas or extracts is undetectable using these classical approaches. Despite advances in fraud detection techniques, adulteration with synthetic safranal has not been well studied to our knowledge. Few preliminary investigations have been undertaken using stable isotope analyses [26,27]. The δ^13^C isotopic analysis of safranal by GC-C-IRMS has already been proposed to determine the naturality of safranal, with synthetic safranal having a δ^13^C slightly different from natural ones [28]. However, this method was not sensitive enough to differentiate total to partial additions and was conducted on an insufficient number of saffron samples that had been tested.

However, the frauds of saffron geographical origins are also of major concern, and the need for methods to authenticate its origin is growing. Stable isotope techniques have demonstrated huge advances in geographical origin authentication. The multi-isotope approach, such as δ^2^H, δ^13^C and δ^15^N determinations undertaken on the bulk saffron [29], permitted one to distinguish the geographical origin of a set of saffron from Italy, Spain, Iran, and Greece; the authors also suggested taking into different years of production and other major producers. When combining analyses of isotopically stable ratios with inorganic trace elements, differences between Iranian and Spanish origins have been highlighted [26]. Geographical classification of Italian saffron has also been studied by HPLC analysis data and linear discrimination [30].

To our knowledge, the analysis of δ^2^H on saffron samples to detect the addition of synthetic safranal has never been published. Moreover, the combination of metabolite quantification through the LC separation technique and the safranal-based stable isotope ratio has never been used to differentiate the saffron origin and its naturality.

Thus, this study aims to propose an authentication method to determine the naturality of safranal contained in saffron samples using stable isotope ratios of safranal and to discriminate the Iranian origin of saffron from others using a multivariate analysis of saffron metabolites and stable isotope ratios of safranal.

## 2. Results

### 2.1. Naturality of Safranal

The dispersion of natural safranal in 33 saffrons and synthetic safranal from 4 available batches, according to δ^2^H and δ^13^C ratios, were very different (Figure 1). The δ^2^H ratio for the synthetic safranal ranged from −19 to 73‰, while the δ^2^H ratio for the natural safranal ranged from −260 to −149‰. The mean δ^2^H ratio was −210‰ (+/− 35) for natural safranal and 36‰ (+/− 40) for synthetic safranal. The δ^13^C ratio for natural safranal ranged between −27‰ and −33‰, the mean ratio was −30‰ (+/− 2), and the δ^13^C ratio for synthetic safranal ranged from 26‰ to −28‰, with a mean ratio of −27‰ (+/− 1).

After quantification of safranal content in Iranian genuine saffron, the spice was voluntarily adulterated with addition of synthetic safranal. Analyses of these fraudulent examples containing 0.25% to 1% synthetic safranal on total safranal showed a negative value of δ^2^H ratio (Table 1), indicating the possibility of detecting small additions of synthetic safranal in natural saffron. The δ^2^H ratio of the adulterated saffron with 1% (synthetic safranal/total safranal) is higher, giving −89, compared to values observed for natural safranal, ranging from −260 to −149 in Figure 1. The addition of 1% synthetic safranal in Iranian saffron was detectable.

### 2.2. Comparison of Stable Isotope Ratios from Safranal and Saffron Metabolites between Countries

Forty-one saffron samples from different geographical origins were analyzed: Iran (N = 20), Spain (N = 9), France (N = 6), Greece (N = 3), Morocco (N = 2), and Italy (N = 1). 

The metabolites content and the δ^13^C of safranal were significantly different from the different saffron geographical origins, based on the results of the Kruskal–Wallis test with post hoc comparisons (Table 2). 

Compared to Spanish and other origins, Iranian saffron had higher δ^13^C ratios of safranal, total crocins, C3, C5, C8, C11, C12, HTTC and trans crocetin and lower level of flavonoid F1 (*p* < 0.05). Compared to Iranian saffron, the Spanish saffron had higher levels of picrocrocin and picrocrocin derivatives (*p* < 0.05) but lower levels of picrocrocin P1 and crocin C2, compared to Iranian or other saffron.

### 2.3. Explorative PCA before Geographical Discrimination

The first three principal components (PC) represented 79% of the total variance from the original data (Figure 2). PC1 provided the greatest variance (45%) and was highly correlated with total and individual crocins (C2, C3, C5, C8, C12, trans crocetin) and HTCC. PC2 represents 21% of the variance and was highly correlated with picrocrocin and picrocrocin derivatives. Finally, PC3, represented by the lowest variance (11%), was strongly correlated with crocin C12 and inversely correlated with crocin C11. From the PCA score plots, plan PC1–PC2 clustered two clear-cut groups, corresponding to Spanish and Iranian saffron. 

PC1 mainly differentiated the Iranian saffron, rich in crocins from the Spanish saffron. The crocin concentrations are mostly determined by postharvest processing, as opposed to the actual geographic origin of the spice, as already mentioned in the literature [26]. However, five saffron samples (one Spanish, two French, one Greek and one Moroccan) were present in the cluster of Iranian samples.

### 2.4. Linear and Quadratic Discriminant Analyses

Two models of discriminant analyses were generated and compared to classify saffron samples based on their saffron metabolites and δ^13^C ratio.

#### 2.4.1. Iran Versus Other Origins

The backward stepwise method selected a relatively small number of variables with high discriminant power to differentiate saffron from Iran from other areas. Variables by order of significance in the stepwise QDA included trans-crocetin, δ^13^C ratio of safranal, picrocrocin, and crocin C3. The QDA model based on the above four selected variables exhibited a very good performance in calibration and prediction (Table 3). The inspection of these data reveals that all the calibration samples were correctly classified. Under the cross-validation conditions, 85% of Iran samples and 95% of other samples were correctly classified.

#### 2.4.2. Iran and Spain Versus Other Origins

The stepwise method selected a relatively small number of variables with high discriminant power to differentiate saffron from three geographical origins. Variables by order of significance in the stepwise QDA included trans-crocetin, crocin C2, δ^13^C, crocin C3 and picrocrocin. The QDA model based on the above six selected variables exhibited a relatively good performance in calibration and prediction (Table 4). The inspection of these data revealed that all the calibration samples were correctly classified. The prediction ability following the cross-validation procedure was 82.5%, 85% of Iranian samples, 100% of Spain samples, and 50% of other samples were correctly classified. This method takes into account an isotopic variable, which allows us to consider not only crocin concentration data but also δ^13^C, which is known to be stable and dependent on pedoclimatic conditions.

## 3. Discussion

To our knowledge, the difference between δ^2^H of natural safranal and δ^2^H of synthetic safranal is observed for the first time with a mean value at −210‰ (+/− 35) for natural safranal and 36‰ (+/− 40) for synthetic safranal. These differences are consistent with another study focused on the authentication of bitter almond oil terpenes [31]. In this cited study, the δ^2^H mean value of natural benzaldehyde was −125‰ and significantly lower than synthetic ones, which were −40‰ and 777‰ for synthetic benzaldehyde obtained from benzal chloride or toluene synthetic pathway, respectively. In another study focused on vanillin authentication, the isotopic ratio from natural vanillin, which has been determined through solid-phase microextraction and gas chromatography–isotope, gave a negative δ^2^H between −91‰ and −75‰, while synthetic vanillin gave a positive δ^2^H between 38‰ and 104‰ [27]. In addition, a synthetic standard of organosulfides showed more positive δ^2^H values than natural standards from Allium species in another recent study [32]. Unlike other aromatic compounds, the availability of synthetic safranal is currently quite limited but it would be interesting to complete the isotopic analyses of safranal obtained with different synthesis routes or different origins and more production batches.

Differences in δ^13^C between Iranian, Spanish, and other saffron show a first level of geographical differentiation with median values of −28.62‰, −30.12‰, and −30.70‰, respectively. No difference in the δ^2^H ratio was observed according to the saffron origin. In another study, the δ^13^C ratio between Spanish and Iranian saffrons were not statistically differents, contrary to δ^2^H ratio for which a higher deuterium content was observed in Iranian saffron [29]. These analyses were performed in the entire saffron, contrary to our present study focused on safranal. It would be interesting to study if an isotopic enrichment can take place after harvesting on specific compounds, such as safranal, or if safranal contains different isotopic ratios than other compounds.

Concerning the chemical composition of the different types of saffron determined by UHPLC-DAD, significant differences were observed between the Iranian and Spanish saffron and saffron from other origins. In order to limit the biases related to the degradation of the crocins and the other compounds of the saffron, the analyses were realized a few months after being harvested.

## 4. Materials and Methods

### 4.1. Chemicals

High purity gases carbon dioxide 99.998% (Messer France SAS, Saint -Georges-d’Espéranche, France) and hydrogen 99.999% (Messer France SAS, Saint -Georges-d’Espéranche, France) were used as working reference gas. Helium flow at 99.999% was used as carrier gas. One microliter of paraffin 1-decane (Sigma Aldrich Corp, St Louis, MO, USA) was injected to install a coating in the pyrolysis alumina tube prior to the start of the analysis process.

International standards IAEA CH7 (polyethylene foil: δ^13^C (‰) −32.15, δ^2^H (‰) −100.3) and IAEA NBS 22 (oil: δ^13^C (‰) −29.79, δ^2^H (‰) −120) supplied by IAEA (International Agency of Atomic Energy) permit the calibration of safranal synthesis standards.

Safranal was purchased from Sigma Aldrich Corp (St Louis, MO, USA). Unfortunately, Sigma Aldrich was the only supplier of synthesis safranal origin found on commercial market. For this reason, three safranal batches were bought at different times, with high levels of purity (98.4%), and were stored at 4 °C., and β-cyclocitral was purchased from Sigma Aldrich (Ref 16976. Purity ≥ 97%); crocin (trans-crocetin-4-GG) was purchased from Phytolab (Ref 80391, purity 99.04%); kaempferol-3-O-glucoside was purchased from Extrasynthese (Ref 1243 S batch purity 99%); trans-crocetin was purchased from Toronto Research (Ref C794950, purity 95%).

### 4.2. Saffron Samples

Thirty authentic dried saffron spices were collected from different producers and cooperatives in Iran (Khorasan province), Spain (AOP Castilla la Mancha), Greece (AOP Krokos Kosanis), Morocco (AOP Taliouine) France (PGI Quercy province), and Italy (Sardegna), with all of them meeting ISO3632 quality criteria and traceability requirements, including certificate of origin obtained from independent laboratories. The saffron crop years were comprised between 2011 and 2019.

### 4.3. Adultered Saffron Samples

For the naturality study, adulterated samples were prepared by adding different amounts of synthetic safranal into a crushed Iranian saffron, from 0.25% to 1% was added to the total safranal contained in saffron (determined by UHPLC-DAD analysis). The addition of 0.5 to 2 µL of pure synthetic safranal was performed in 1 mL of a preparation containing authentic saffron powder and methanol (10% *w*/*v*). Methanol was chosen for its ability to solubilize safranal.

### 4.4. UHPLC-DAD Analysis

The quantification of saffron metabolites (crocins, picrocrocin and its derivatives, safranal, kaempferol derivatives) was analyzed according to the UHPLC method, as previously described [24]. The UHPLC system used was an UltiMate 3000 RS with autosampler RSCL WPS-3000RS coupled with DAD-3000 (Thermo Fisher Scientific, Courtaboeuf, France).

One hundred milligrams of crushed saffron stigma were extracted by 19 mL of MeOH 50% in distilled water in a 20 mL brown glass flask, consisting of 5 min sonication at 30 °C followed by 1 h of magnetic stirring at room temperature and 5 min of sonication at 30 °C. After filling the flask, the product was slightly mixed and diluted in MeOH 50% in water before filtration through a 0.45 µm PTFE membrane. In other recent studies, MeOH 50% was the most suitable solvent to extract a wide range of compounds, including crocins and safranal, among different solvents: water, ethanol, methanol, ethanol:water (1:1), and methanol:water (1:1) [33].

LC separation was performed through a core–shell silica reversed-phase, Kinetex C18, Phenomenex 2.6 μm, 150 × 2.1 mm. The mobile phase was 0.01% formic acid in water (A) and acetonitrile (B). The elution and calibration were performed, as mentioned in the study. The results are presented in absolute concentration, the quantification is obtained exactly according to the calculation mode described in our previous work [24].

The calibration curves are obtained with:$${\left[Reference\;compound\right]}_{mg/L}=a\;Peak\;Area+b$$

All compounds are fitted to the coefficients of the linear equation (slope and intercept), obtained with each corresponding standard as follows:$${\left[Compound\right]}_{mg/L}=a\;Peak\;Area+b$$

All results are express in % of powder (*w*/*w*) as follows:$$\%\;Compound=\frac{{\left[Compound\right]}_{mg/L}\times V}{w}$$
where *V* is the volume of the sample preparation and *w* the sample weight of the preparation. Total crocins are expressed as Crocin trans-4-GG. The content of Picrocrocine or derivatives are expressed as β-cyclocitral equivalent. The content of kaempferol derivatives is expressed as kaempferol glucoside equivalent. Safranal is quantified with its proper standard. For picrocrocin, a conversion factor was used to convert the β-cyclocitral equivalent expression into picrocrocin according to their respective molecular weights (i.e., conversion factor = 2.17).

Validation criteria have been determined according to ICH Guidance, the limit of detection was 0.012 mg/L, 0.024 mg/L, 0.043 mg/L and 0.012 mg/L, respectively, for crocins (trans-4-GG Eq.), safranal, Kaempferol 3-*O*-sophoroside, picrocrocin (beta-cyclocitral Eq.). The limit of quantification was, respectively 0.037 mg/L, 0.072 mg/L, 0.1304 mg/L and 0.037 mg/L.

Linearity was determined from 0.10 mg/L to 225.29 mg/L for trans-4-GG giving coefficient of determination (R2) of 0.9988; from 0.5 mg/L to 500 mg/L for the safranal and β-cyclocitral giving R2 of 0.9997 and 0.9998; and from 0.10 mg/L to 509 mg/L for the Kaempferol 3-*O*-sophoroside.

### 4.5. Isotopic Analysis

#### 4.5.1. EA-IRMS Analysis

Both ^13^C and ^2^H isotopic ratio of synthesis safranal used as reference standards for the measurements of saffron using GC-C/P-IRMS (TRACE 1310 GC-Isolink II-Delta V Plus IRMS, Thermo Fisher Scientific, Bremen, Germany)) were first determined using Bulk stable isotope analysis on 2 EA-IRMS coupling systems (Flash HT EA-Conflo IV interface- Delta V Plus IRMS, Thermo Fisher Scientific, Bremen, Germany) and (TCEA-Conflo IV interface-Delta V Advantage IRMS, Thermo Fisher Scientific, Bremen, Germany).

The carbon isotopic analysis (δ^13^C value) of synthetic safranal standards was measured using a Flash HT elemental analyzer connected to a Delta V Plus via a Conflo IV interface (all Thermo Fisher Scientific). Samples in tin capsules (5 × 8, Säntis Analytical AG, Teufen, Switzerland) fell in the unit combustion maintained at 950 °C in a stream of 100 mL min^−1^ helium with a pulse of oxygen. Flash combustion (close to 1800 °C) allowed the complete oxidation of carbon into CO_2_. The gases conveyed throughout a reduction furnace held at 650 °C were made by copper wires, where excess oxygen was trapped, followed by a water trap containing anhydrous magnesium perchlorate. A Flash HT elemental analyzer was connected to a Thermo Delta V Plus isotope ratio mass spectrometer via an online interface (Conflo IV). This intermediary device permitted the adjustment of automatic sample dilution and the generation of reference gas pulses, allowing isotopic determination.

The δ^2^H measurements of synthetic safranal standards were carried out using a TC-EA elemental analyzer linked to a Delta V Advantage via a Conflo IV interface (all Thermo Fisher Scientific). Samples weighed in silver capsules (3.3 × 5 mm, Säntis Analytical AG, Teufen, Switzerland) dropped in the pyrolysis unit were held at 1450 °C. This high-temperature reactor was made of an inner glassy carbon tube, half filled with glassy granular carbon, integrated into an external ceramic tube. Hydrogen and oxygen were converted in H_2_ and CO and separated on a gas chromatography column prior to being introduced in the IRMS. The interface operated similarly to that previously described, providing steady and regular reference gas pulses.

The isotope ratio of a sample is typically expressed in δ notation according to the following formula, defined as:(1)$$\delta \mathrm{iC}\;=\left(\mathrm{iRA}/\mathrm{iRstd}-1\right)$$
where ‘i’ is the isotope ^2^H or ^13^C and iR correspond to ^2^H/^1^H and ^13^C/^12^C ratio, respectively.

The delta values were multiplied by 1000 and expressed in units “per mil” (‰).

δ^13^C and δ^2^H were determined versus the Vienna Pee Dee Bee Belemnite (VPDB) ^13^C international standard and standard mean ocean water (SMOW) ^2^H international standard. 

Five replicates of synthetic safranal were done to determine with a good precision δ^13^C and δ^2^H values. The uncertainty was assessed at +/− 0.30 (‰) for ^13^C and +/− 3 (‰) for ^2^H.

#### 4.5.2. Extraction Procedure

1 g of saffron sample was suspended in 10 mL methanol and placed in an ultrasound bath for 30 min. After 10 min of decantation time, the liquid supernatant was recovered and concentrated to 1 mL by evaporation under nitrogen 20 mL/min flow.

#### 4.5.3. GC-C/P-IRMS Analysis

GC-C/P-IRMS was performed using a GC Trace 1310 doubly connected to a single quadrupole mass spectrometer ISQ and an isotope ratio mass spectrometer Delta V Plus via a GC Isolink II interface for δ^13^C and δ^2^H measurements. (Thermo Fisher, Bremen, Germany). This equipment enables the identification of peaks and isotopic ratio measurements in the same runoff injection.

From 1 to 3 µL of liquid sampling were injected using a Tri Plus RSH autosampler in a TRACE 1310 Gas Chromatograph equipped with an Agilent DB-XLB column (30 m × 0.25 mm, 0.25 µm I. D film thickness) for chromatographic separation. The injection port was held at 250 °C, fitted with a split liner containing glass wool and operated in split mode of (1:10). The oven temperature program started at 50 °C for 2 min, increased to 125 °C at a rate of 10 °C min^−1^, 170 °C at a rate of 5 °C min^−1^ and 340 °C at a rate of 20 °C min^−1^ (held for 5 min at this temperature). Components were separated at a flow rate of 1 mL min^−1^ and divided in two ways using a Sil Flow connector.

The combustion of carbon in carbon dioxide was undertaken using an alumina tube containing nickel oxide and held at 1000 °C in a furnace. Pyrolysis of the sample for ^2^H measurement in H_2_ was carried out at higher temperature (1400 °C) using a thermo-conversion tube made by an empty alumina tube, where a layer of carbon was deposited by injections of 1 decane.

The system was checked by passing a synthetic safranal standard sample every two analyses. Each sample was determined in two replicates. The uncertainty was established at +/− 0.50 (‰) for ^13^C and +/− 5(‰) for ^2^H.

### 4.6. Statistical Analyses

Statistical analysis was conducted using SAS software version 9.4 (SAS Institute, Cary, NC, USA). Saffron metabolites with more than 20% of values below the detection limit (DL) were removed. For saffron metabolites with less than 20% of values below the DL, any values reported as <DL were substituted with half the value of the DL. This strategy was used to avoid introducing bias through the substitution of a many censored values, while trying to minimize loss of information through the removal of variables [26].

Normality of the saffron metabolites and the two stable isotope ratios were tested using the Shapiro-Wilk test. As several metabolites were not normally distributed, comparisons according to three geographical origins (Iran, Spain, and other origins) were performed with the Kruskal-Wallis test. Dwass, Steel, Critchlow-Fligner multiple comparisons post hoc procedure was done to determine which pairs significantly differed. Differences were considered statistically significant for *p* < 0.05.

Prior to multivariate statistical analysis, saffron metabolites and isotope ratios were standardized to z scores to impart normality on the dataset and reduce the appearance of outliers. Principal component analysis (PCA) was performed on the transformed variables, and the percentage of variance for each specific principal component was reported.

Classificatory discriminant analysis was used to differentiate the saffron samples from different geographical origins (Iran, Spain, and other areas) based on stable isotope ratios and saffron metabolites. The discriminant variables were identified by a backward or stepwise algorithm based on Wilks’ lambda method using PROC STEPDISC. When the within-class covariance matrices were assumed to be unequal, a quadratic discriminant function (QDA) was preferred to a linear discriminant function (LDA). This supervised multivariate classification technique determines combinations of variables to maximize between-group variance while minimizing within-group variance. QDA predictive ability was evaluated by the leave-one-out cross-validation method

## 5. Conclusions

The stable isotope ratios of carbon and hydrogen from safranal may be a useful tool for the authentication of saffron. The δ13C between Iranian, Spanish and other saffron was significantly different and represents median values of −28.62‰, −30.12‰ and −30.70‰, respectively. The δ2H ratio allows the controlling of the absence of the synthetic safranal in significant proportion to identify a possible adulteration of saffron. An addition of 1% synthetic safranal to total safranal could be detected in the adulterated samples. In addition, the UHPLC-DAD analysis of crocins, picrocrocin derivatives, kaempferol, and safranal coupled with the analysis of the stable isotope δ13C ratio of safranal allows us to distinguish Iranian saffron from other origins. Further investigation is required to extend these models on saffron samples from other major producers, such as Spain, Morocco, or Greece.

## Figures and Tables

**Figure 1 molecules-27-06801-f001:**
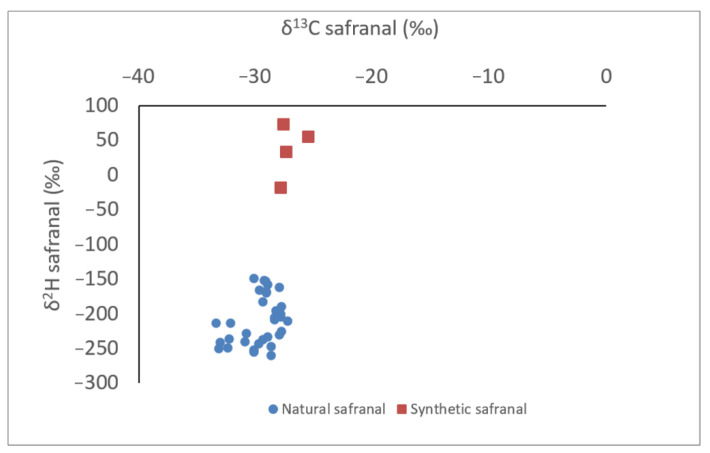
Representation of natural safranal and synthetic safranal according to δ^2^H and δ^13^C ratios.

**Figure 2 molecules-27-06801-f002:**
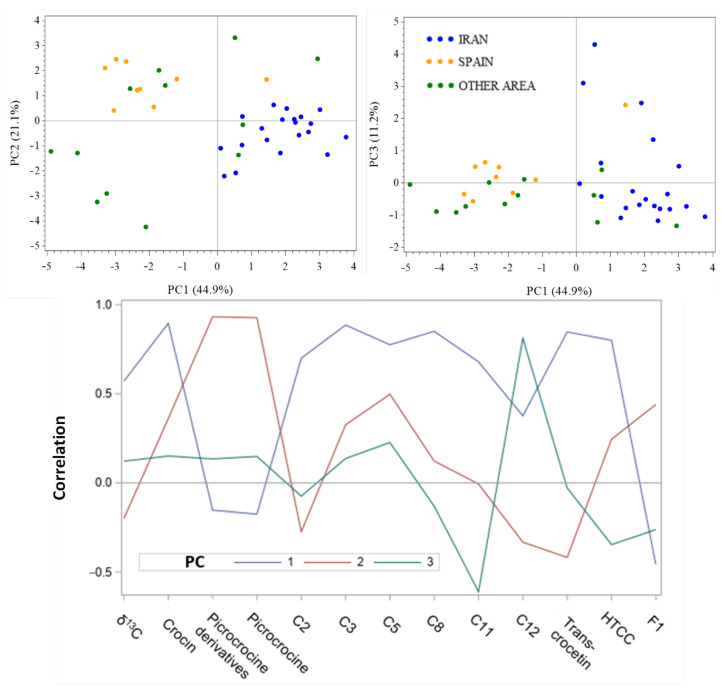
Saffron samples projected in the space of the 3 PC (**above**) and PC pattern profile (**below**).

**Table 1 molecules-27-06801-t001:** Isotope analysis of adulterated Iranian saffron by synthetic safranal (mean of five analyses).

Synthetic/Natural Safranal	0	0.25%	0.5%	0.75%	1%
δ^2^H ratio	−221 ± 3	−172 ± 4	−126 ± 4	−110 ± 3	−89 ± 5
δ^13^C ratio	−26.19 ± 0.11	−25.98 ± 0.12	−25.79 ± 0.08	−25.69 ± 0.10	−25.52 ± 0.09

**Table 2 molecules-27-06801-t002:** Comparison of safranal isotope ratios (‰) and saffron metabolites (% *w*/*w*) included in saffron samples according to geographical origins (N = 40).

	Iran (N = 20)		Spain (N = 9)		Other Area (N = 12)		
	Median	IQ Range	Median	IQ Range	Median	IQ Range	*p* Value *
δ^13^C	−28.62 ^a,b^	0.90	−30.12 ^a^	2.74	−30.70 ^b^	2.80	0.0002
δ^2^H	−227.75	25.25	−236.50	9.0	−236.50	31.50	0.8
Total crocin	20.61 ^a,b^	2.61	15.60 ^a^	0.98	16.28 ^b^	7.91	0.0004
Crocin C3 *	11.31 ^a,b^	1.07	8.13 ^a^	0.87	9.21 ^b^	4.73	0.0001
Picrocrocin derivative	8.10 ^a^	1.10	10.80 ^a^	1.82	7.81	4.27	0.001
Picrocrocin	7.84 ^a^	1.05	10.66 ^a^	1.83	7.73	4.34	0.0009
Crocin C5 *	4.98 ^a^	0.75	4.18 ^a^	0.73	4.55	2.52	0.006
Crocin C8 *	1.82 ^a,b^	0.18	1.29 ^a^	0.22	1.28 ^b^	0.62	0.002
Crocin C7 *	1.01	0.21	1.12	0.41	0.77	0.40	0.02
Kaempferol derivative	0.97	0.12	0.93	0.12	0.88	0.09	0.1
Flavonoid F5 *	0.67 ^a^	0.06	0.56	0.18	0.49 ^a^	0.13	0.02
Flavonoid F1 *	0.29 ^a,b^	0.06	0.33 ^a^	0.07	0.36 ^b^	0.11	0.0005
HTCC	0.31 ^a,b^	0.08	0.13 ^a^	0.07	0.17 ^b^	0.15	0.001
Crocin C1 *	0.22	0.05	0.21	0.04	0.19	0.09	0.7
Safranal	0.17	0.08	0.18	0.03	0.14	0.11	0.6
Crocin C6 *	0.23	0.08	0.23	0.15	0.12	0.15	0.04
Crocin C11 *	0.17 ^a,b^	0.07	0.10 ^a^	0.03	0.12 ^b^	0.04	0.007
Crocin C10 *	0.03	0.04	0.01	0.01	0.02	0.01	0.2
Crocin C4 *	0.11	0.14	0.13	0.02	0.10	0.09	0.3
Crocin C2 *	0.10 ^a^	0.03	0.02 ^a,b^	0.01	0.07 ^b^	0.06	<0.0001
Crocin C13 *	0.05	0.02	0.05	0.02	0.05	0.04	0.4
Crocin C9 *	0.02	0.06	0.01	0.01	0.01	0.04	0.4
Crocin C12 *	0.04 ^a,b^	0.08	0.01 ^a^	0.01	0.02 ^b^	0.02	0.0001
Crocin C14 *	0.04	0.02	0.04	0.02	0.03	0.03	0.3
Picrocrocin P1 *	0.03 ^a^	0.01	0.01 ^a,b^	0.00	0.02 ^b^	0.04	0.001
*trans*-Crocetin	0.03 ^a,b^	0.00	0.01 ^a^	0.00	0.01 ^b^	0.01	<0.0001

*p* values for Kruskal–Wallis test. Dwass, Steel, Critchlow–Fligner multiple comparisons post hoc procedure was done to determine which pairs differ: significant pairs differences were indicated with ^a,b^. * The compound nomenclature is in accordance with previous work [24].

**Table 3 molecules-27-06801-t003:** Confusion matrices displaying the classification performance of QDA applied to saffron extract samples from two geographical origins.

	Classification Matrix			Prediction Matrix	
	**Assigned Class**			**Assigned Class**	
True class		Iran	Other area	Iran	Other area
	Iran	20	0	17	3
	Other area	0	21	1	20
% of correct classification		100	100	85	95
Total ability 90.24%					

**Table 4 molecules-27-06801-t004:** Confusion matrices displaying the classification performance of QDA applied to saffron extract samples from three geographical origins.

	Classification Matrix			
	Assigned Class			
True class		Iran	Spain	Other area
	Iran	20	0	0
	Spain	0	8	0
	Other area	0	0	12
% of correct classification Total ability: 100%		100	100	100
	Prediction matrix			
	Assigned class			
True class		Iran	Spain	Other area
	Iran	17	0	3
	Spain	0	4	4
	Other area	0	0	12
% of correct classification		85	100	50
Total ability: 82.5%				

## Data Availability

Not applicable.

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
