# Peer review of "Authentication of Iranian Saffron (Crocus sativus) Using Stable Isotopes δ13C and δ2H and Metabolites Quantification"

_molecules, 2022, doi:10.3390/molecules27206801_

Round 1

Reviewer 1 Report

The present work deals with the authentication of Iranian saffron (Crocus sativus) using stable

isotopes δ13C and δ2H and metabolites quantification. Saffrons from different countries were analysed to determine the stable isotope ratios δ13C and δ2H from safranal by GC coupled with IRMS and the concentration of saffron metabolites with UHPLC-DAD. The isotopic analysis highlighted higher ratio of δ2H in synthetic safranal than in natural safranal.

The work is interesting and well-conducted. In fact, saffron which is a very high value-added ingredient used in the food supplement market, is often the object of fraud. The combined analytical approach investigated in this study may be a useful tool to authenticate the origin of unknown saffron.

However, important amendments are necessary prior to eventual publication.

-          English needs an extensive polishing e.g. Abstract, lines 18-19: “were significantly different”.

-          Line 19: The acronym C/P is not reported.

-          Line 28 and elsewhere: “Crocus sativus” must be italized.  

-          Line 136 and elsewhere: 13 must be superscript.  

-          Line 142: Erase o after %“(‰)

-          Lines 212-216. This part is too maigre. The way how the metabolite content was achieved must be reports in much more critical way. Also, a comparison with previously published data must be reported.

-          Line 219 and elsewhere: 2 must be superscript.  

-          Heading of Table 2: Avoid bold.

-          Lines 266-267: The UHPLC-DAD system have to be described.

-          Lines 267-274 should be erased from there and placed in a new sub-section.

-          Lines 280-284: the content reported is obvious and must be erased.

Author Response

Thank you for accepting to review this manuscript and for your relevant comments.

-          English needs an extensive polishing e.g. Abstract, lines 18-19: “were significantly different”.

We have corrected English language. For your information, we have requested the correction of the article by an online service of Elsevier, a scientific journal editor . However, it is true that there were still some errors, we have corrected them following your comment.

Line 19: The acronym C/P is not reported.

yes indeed, we modify it, thank you

-          Line 28 and elsewhere: “Crocus sativus” must be italized.

We modified it, thank you

-          Line 136 and elsewhere: 13 must be superscript.

We modified it, thank you

-          Line 142: Erase o after %“(‰)

Sorry I didn’t find a “%” or “‰” on line 142 (or around).

-          Lines 212-216. This part is too maigre. The way how the metabolite content was achieved must be reports in much more critical way. Also, a comparison with previously published data must be reported.

This part, line 212-216, is exclusively dedicated to a bibliographic reference work which explains that other studies go in the same direction as our results. We have cited 3 references and described their work. As I deduced that this part is not well detailed, please find the following suggestion:

“To our knowledge, the difference between δ2H of natural safranal and δ2H of synthetic safranal is observed for the first time with a mean value at -210 ‰ (+/- 35) for natural safranal and 36 ‰ (+/- 40) for synthetic safranal. These differences are consistent with the results of another study focused on the authentication of bitter almond oil terpens (Butzenlechner et al., 1989). In this cited study, the δ2H mean value of natural benzaldehyde was -125 ‰ which was significantly lower than the synthetic ones which were -40 ‰ and 777 ‰ for synthetic benzaldehyde obtained from benzal chloride or toluene synthetic pathway respectively. In another study focused on vanillin authentication, the isotopic ratio from natural vanillin, which has been determined through solid-phase microextraction and gas chromatography-isotope, gave a negative value of δ2H ranging from -91‰ to -75‰, while synthetic vanillin gave a positive value of δ2H ranging from  38‰ to 104‰ (Perini et al., 2019). In addition, a synthetic standard of organosulfides showed more positive δ2H values than natural standards from Allium species in another recent study (Cuchet et al., 2021). »

-          Line 219 and elsewhere: 2 must be superscript. 

We modified it, thank you

-          Heading of Table 2: Avoid bold.

I removed bold police from all headings more in line with the practices, thank you

-         

- Lines 266-267: The UHPLC-DAD system have to be described.

Yes, this analytical system is the same as the cited article (Moras et al., 2018). I added the following description: “The UHPLC system used was an UltiMate 3000 RS with autosampler RSCL WPS-3000RS coupled with DAD-3000 (ThermoFisher Scientific).”

-          Lines 267-274 should be erased from there and placed in a new sub-section.

Yes, I created a new section intitled “4.3.          Adultered saffron samples”, thank you

-          Lines 280-284: the content reported is obvious and must be erased.

In order to avoid any ambiguity, we preferred detailing this part, but we appreciate your recommendations to improve our work, so we remove it, thank you.

Reviewer 2 Report

The analysis of δ2H on saffron samples to detect the addition of synthetic safranal in this paper is very interesting, and it is very important to the authentication of the raw materials or extracts.

But the paper only give the analysis result of δ2H and δ13C ratio of different samples. What is the main chemical structure of safranal ? and which function group lead to the changes ofδ2H and δ13C ratio?

Author Response

Thank you for your very encouraging comments that recognizes the work done.

The ways of obtaining the synthetic safranal used in our study are not precisely known. In addition, there is very little evidence in the literature on this subject. The existing synthesis pathways consist of dehydrogenation or bromination of cyclocitral, or synthesize safranal from the enol acetate of cyclocitral (http://hdl.handle.net/1721.1/60741). Another reaction consists of cyclizing olefinic β-hydroxy selenide (Kametani et al., 1979). Unfortunately, we do not know if the safranal produced is obtained one of these ways. Ideally, we should study different ways of synthesis of safranal and examine the impact on the isotopic ratios (for example the enrichment in deuterium).

Given this lack of available information, we think it seems perilous to make assumptions about the potential enrichment in deuterium for example. I hope to convince you.

Reviewer 3 Report

Comments and Suggestions for Authors

Manuscript ID: molecules-1928117

Comments:

In the manuscript entitled “Authentication of Iranian saffron (Crocus sativus) using stable isotopes δ13C and δ2H and metabolites quantification, by : Benjamin Moras, Camille Pouchieu, David Gaudout , Stéphane Rey, Anthony Anchisi, Xavier Saupin, Patrick Jame, reported the analysis of Saffrons from different countries to determine the stable isotope ratios δ13C and δ2H from safranal by Gaz Chromatography coupled with Isotope-Ratio Mass Spectrometry (GC-C/P-IRMS) and the concentration of saffron metabolites with Ultra High-Performance Liquid Chromatography coupled with Diode Array Detector (UHPLC-DAD). They described  combined approach to authenticate Iranian saffron. The results obtained A first QDA showed that trans-crocetin and the δ13C of safranal, picrocrocin and crocin C3 concentrations clearly differentiated Iranian saffron from other origins. A second model identified trans-crocetin, crocin C2, δ13C, crocin C3 and picrocrocin as good predictors to discriminate saffron samples from Iran, Spain or other origins.

The study is well designed, the methods used to validate the results are adequate and accurate. The introduction provides sufficient background, and some additional  references are needed. The results are clearly presented with good tables with statistical analyses and figures with good resolution (Can be improved). The conclusion section is supported by the validation of the two techniques propped to authenticate saffron from Iran in comparison to Spain origin (especially).                  

However, there are some modifications required to be done before it is accepted for publication. The following are the specific comments on the manuscript:

Specific comments:

  1. Please add some numbers in the Abstract and Conclusion parts to make them more scientifically sound
  2. Many paragraphs are two long and need additional references. Add reference for Line 34-35. Add reference line 68-70, Add reference Line 86-88.
  3. Crocus sativus: have to be in italic in the whole manuscript, please use C. sativus after the first citation.
  4. Figure 1: please provide a better picture with high resolution

Author Response

Thank you for accepting to review this manuscript and for your relevant comments.

  1. Please add some numbers in the Abstract and Conclusion parts to make them more scientifically sound

We added some values accordingly:

“The isotopic analysis highlighted higher ratio of δ2H in synthetic safranal than in natural safranal, the mean values were 36 ‰ (+/- 40) and -210 ‰ (+/- 35) respectively. The δ13C between Iranian, Spanish and other saffrons were significantly different and represent median values of -28.62 ‰, -30.12 ‰, and -30.70 ‰ respectively.”

“A second model identified trans-crocetin, crocin C2, δ13C, crocin C3 and picrocrocin as good predictors to discriminate saffron samples from Iran, Spain or other origins with a total ability score classification matrix of 100% and a prediction matrix of 82.5%.”

We add the δ13C in the discussion/conclusion part.

Thank you

2. Many paragraphs are two long and need additional references. Add reference for Line 34-35. Add reference line 68-70, Add reference Line 86-88.

We understand, thank you, we have shortened some paragraphs and as requested, we added some additional references:

Line 34-35

Iran is currently the main producer, with approximately 90% of world production which was estimated at 418 tons in 2020 (Cardone et al., 2020). The other main producers are India, Greece, Afghanistan, Morocco, Spain, Italy and China (Shahnoushi et al., 2020). Spain is the first saffron importer. Export and import trading data comparing to production showed that Spain creates more value-added products in the saffron industry than other countries (Mohammadi and Reed, 2020). Saffron from Iran could be traditionally declined in different grades, the Sargol grade corresponds to the top of the red stigma, which allows having a high content in crocins compounds that gives the saffron its characteristic color (Jafari et al., 2020).

  • [3] Shahnoushi, N.; Abolhassani, L.; Kavakebi, V.; Reed, M.; Saghaian, S. Chapter 21 - Economic Analysis of Saffron Production. In Saffron; Koocheki, A., Khajeh-Hosseini, M., Eds.; Woodhead Publishing, 2020; pp. 337–356 ISBN 978-0-12-818638-1.
  • [5] Jafari, S.-M.; Tsimidou, M.Z.; Rajabi, H.; Kyriakoudi, A. Chapter 16 - Bioactive Ingredients of Saffron: Extraction, Analysis, Applications. In Saffron; Koocheki, A., Khajeh-Hosseini, M., Eds.; Woodhead Publishing, 2020; pp. 261–290 ISBN 978-0-12-818638-1.

For the line 68-70

  • Koocheki, A.; Milani, E. Chapter 20 - Saffron Adulteration. In Saffron; Koocheki, A., Khajeh-Hosseini, M., Eds.; Woodhead Publishing, 2020; pp. 321–334 ISBN 978-0-12-818638-1

For the line 86-88

We already added the following reference [25]:

  • Semiond, D.; Dautraix, S.; Desage, M.; Majdalani, R.; Casabianca, H.; Brazier, J.L. Identification and Isotopic 511 Analysis of Safranal from Supercritical Fluid Extraction and Alcoholic Extracts of Saffron. Analytical Letters 1996, 512 29, 1027–1039, doi:10.1080/00032719608001453.

3. Crocus sativus: have to be in italic in the whole manuscript, please use C. sativus after the first citation.

Yes, we changed this format, thank you

4. Figure 1: please provide a better picture with high resolution

 We have enlarged the figure 1 with a higher resolution,

thank you very much

Round 2

Reviewer 1 Report

The authors have adequately addressed all remarks. There is still a stange sign: ‰ at lines 20, 289-293, 329, 401,402, 404, 405, 409, 410, 416, 476, 477, 610, 614, 649, 650  that needs to be replaced with % but it can be fixed during proofreading.